# H-QLoRA: Enhancing Quantized LLMs with Hierarchical Learning

## Abstract

Fine-tuning large language models (LLMs) in resource-constrained environments poses significant challenges due to their size and computational demands. While current methods often rely on aggressive weight quantization to alleviate memory and computational costs, this can lead to a noticeable loss of accuracy. This paper introduces H-QLoRA, a novel approach that leverages hierarchical adaptors with low-rank weights to enhance performance. By fine-tuning models from the LLaMA and Gemma families, we demonstrate H-QLoRA's efficacy across multiple instruction datasets. H-QLoRA not only outperforms state-of-the-art results for certain model types by recovering high-frequency information lost during 4-bit weight quantization, but it also maintains efficiency in terms of inference costs and memory usage. While traditional methods may compromise accuracy in pursuit of efficiency, H-QLoRA mitigates this issue by implementing a hierarchical adaptor structure that captures more nuanced patterns within the data. This allows H-QLoRA to fine-tune models with the same number of trainable parameters as QLoRA, yet it proves to be more optimal for specific architectures. Overall, H-QLoRA aims to enhance fine-tuning outcomes for quantized models in low-resource environments.

## 1 Introduction

Recent advances in Large Language Models (LLMs) have demonstrated exceptional capabilities in text understanding and generation. To fully leverage the potential of these models, many applications require fine-tuning pretrained architectures for specialized tasks. However, as model sizes increase, the cost of full fine-tuning becomes prohibitively high. In this context, Parameter-Efficient Fine-Tuning (PEFT) techniques (Xu et al., 2023), particularly Low-Rank Adaptation (LoRA) (Hu et al., 2022), have emerged as promising alternatives. LoRA effectively freezes pretrained weights and introduces trainable low-rank matrices into the Transformer architecture, significantly reducing the number of parameters that need adjustment. Despite these advantages, fine-tuning large models still requires substantial computational power and GPU memory.

In parallel, significant efforts have been made in the realm of quantization to decrease the memory footprint of large AI models. QLoRA (Dettmers et al., 2023) has enabled memory-efficient fine-tuning by quantizing model weights, allowing for a 65 billion parameter model to be fine-tuned on a single 48GB GPU while retaining full 16-bit performance. However, QLoRA's quantization to NF4, a 4-bit format with a normally distributed data range, can lead to a loss of resolution in high-frequency components, which LoRA adapters aim to recover. Unfortunately, the low-rank nature of the intermediate matrices limits this recovery, and scaling beyond $r = 64$ has proven unmanageable.

To address these resolution challenges, we introduce H-QLoRA: Hierarchical Learning in Low-Rank Adaptation. This novel approach incorporates a hierarchical structure by splitting the QLoRA adapter into multiple specialized layers, each designed to recover lost resolution across varying data frequencies. Notably, our implementation ensures that all layers share a unified objective function, which, while not explicitly encouraging complementary information encoding, still yields strong results. By adopting a targeted loss function that prioritizes high-frequency detail learning at lower levels, we anticipate further performance improvements over QLoRA, while maintaining a consistent parameter count.

H-QLoRA's hierarchical learning framework facilitates improved retention and recovery of high-frequency data, a common issue in quantization, especially during the fine-tuning of adapters. This hierarchical design allows layers to specialize in different frequency ranges, optimizing parameter efficiency and enhancing model adaptability.

We evaluate H-QLoRA by fine-tuning models from the LLaMA (Touvron et al., 2023a;b; Team, 2024) and Gemma (Gemma Team, 2024) families across various instruction datasets and benchmarks, with a particular focus on the OASST1 dataset and performance as measured by MMLU benchmarks. Initial results indicate that H-QLoRA effectively recovers high-frequency data, leading to improved task performance. Specifically, fine-tuning LLaMA 7B, LLaMA-2 7B, and Gemma 2B shows that H-QLoRA not only sustains model performance but also enhances it without increasing inference costs or memory requirements. This hierarchical framework underscores H-QLoRA's capability to efficiently utilize quantized data, yielding superior outcomes.

In summary, our paper presents the following contributions:

- **Novel Hierarchical Quantization Method:** Introducing H-QLoRA, which incorporates hierarchical learning via multi-adaptor training, improving upon QLoRA.

- **Improved Performance:** Demonstrating enhanced performance over QLoRA for certain models by effectively recovering high-frequency data lost during quantization.

- **Efficiency in Resource-Constrained Environments:** Maintaining efficiency in terms of memory and computational demands, making H-QLoRA suitable for deployment in resource-constrained settings.

- **Experimental Validation:** Validating the effectiveness of H-QLoRA through extensive experiments on the LLaMA and Gemma model families, showcasing significant enhancements across diverse datasets and tasks.

## 2 RELATED WORK

**Large Language Models (LLMs):** The evolution of Large Language Models (LLMs) began with GPT-1 (Radford et al., 2018), which was based on the Transformer architecture (Vaswani et al., 2017). This foundational model paved the way for subsequent models, notably GPT-2 (Radford et al., 2019), GPT-3 (Brown et al., 2020), and the most recent GPT-4/GPT-4o(OpenAI, 2024) and GPT o1. Each successive model has markedly improved capabilities in understanding and generating human-like text, showcasing advancements in contextual understanding and coherence. Beyond the GPT series, a diverse range of LLMs has emerged, including Mistral (Jiang et al., 2023), Gemini (Gemini Team, 2023), LLaMA (Touvron et al., 2023a;b), and Gemma (Gemma Team, 2024). These models have demonstrated robust performance across various natural language processing tasks, including question answering, summarization (Stiennon et al., 2020), and entity recognition (Zhao et al., 2023). Notably, the introduction of open-access models like LLaMA has democratized access to powerful LLMs, enabling researchers to explore applications and fine-tuning methodologies without the barriers of resource-intensive training.

Our work contributes to this evolving landscape by developing an efficient algorithm for Parameter-Efficient Fine-Tuning (PEFT). Rather than training LLMs from scratch, we focus on enhancing their performance for downstream tasks while maintaining rapid inference speeds. By leveraging our proposed H-QLoRA, we aim to facilitate the deployment of LLMs in real-world applications where efficiency and responsiveness are paramount.

**Parameter Efficient Fine-Tuning (PEFT):** Parameter Efficient Fine-Tuning (PEFT) has emerged as a pivotal strategy for adapting large language models (LLMs) to specific tasks while minimizing computational costs. Traditional fine-tuning methods often necessitate adjusting all model parameters, which can be resource-intensive and impractical for deployment in constrained environments (Xu et al., 2023). To tackle these challenges, PEFT techniques selectively update a subset of parameters. Early approaches like Layer-wise Adaptive Learning Rates (LALR) (You et al., 2020) enable differential learning rates across layers, effectively preserving deep-layer knowledge while adapting to new tasks. Sparse fine-tuning methods (Ansell et al., 2022; Han et al., 2024) enhance this by learning a mask that identifies critical parameters for updates. Adapters have gained traction in PEFT, introducing small trainable modules into pre-trained models. The Adapter architecture (Houlsby et al.,

2019) allows for modular adjustments with minimal disruption, while AdapterFusion (Pfeiffer et al., 2020) optimizes performance by stacking multiple adapters. Recent innovations, including Parallel Adapters like Adaptformer (Chen et al., 2022), CoDA (Lei et al., 2023), and KronA (Edalati et al., 2022), reorganize adapter layers into parallel configurations to enhance computational efficiency. Selective Parameter Fine-Tuning (SFT) methods refine the fine-tuning process by focusing on specific subsets of parameters based on metrics such as Fisher information (Sung et al., 2021), mask matrices (Guo et al., 2020), or the Lottery Ticket Hypothesis (Frankle & Carbin, 2018). Although effective, these methods often involve complex selection processes that can hinder performance and increase computational demands.

Low-Rank Adaptation (LoRA) (Hu et al., 2021) is particularly notable for injecting task-specific knowledge through low-rank matrices, which drastically reduces the parameters requiring fine-tuning. Dynamic methods like DyLoRA (Valipour et al., 2022) and AdaLoRA (Zhang et al., 2023) optimize rank selection adaptively, minimizing overfitting and enhancing model efficiency without the need for full retraining. A notable advancement in this field is Quantized Low-Rank Adaptation (QLoRA) (Dettmers et al., 2023), which combines quantization with LoRA. Traditional quantization reduces model weights from high-precision formats, such as 32-bit floating point, to lower precision like 8-bit (Dettmers et al., 2022) or even 4-bit integers (Frantar et al., 2023; Dettmers & Zettlemoyer, 2022), thereby significantly reducing the model's memory footprint. By quantizing low-rank matrices, QLoRA enhances memory efficiency and inference speed, making it ideal for resource-constrained environments. This method retains the advantages of LoRA while lessening the computational demands of high-precision calculations.

The combination of PEFT and quantization techniques creates a powerful synergy for adapting LLMs to specific tasks, particularly in real-world applications where computational efficiency is paramount. Our work contributes to this landscape by exploring new avenues for implementing PEFT methods, such as QLoRA, while ensuring that the integrity and performance of the underlying models are preserved. H-QLoRA aims to facilitate more efficient deployments of LLMs, paving the way for their use in diverse and resource-constrained environments.

## 3 METHOD

### 3.1 PRELIMINARY: LoRA AND QLoRA

Low-Rank Adaptation (LoRA) is a powerful technique designed for the efficient fine-tuning of large language models (LLMs). Traditional full fine-tuning methods require modifying and backpropagating through all model parameters, which is computationally expensive and memory-intensive. LoRA innovatively addresses these challenges by incorporating low-rank decomposition into the model's weight matrices.

Instead of updating the entire weight matrix, LoRA freezes the base model's parameters and only updates two low-rank matrices, denoted as $\mathbf{L}_a$ and $\mathbf{L}_b$. This approach significantly reduces the number of trainable parameters and associated computational costs. By leveraging the knowledge captured during the model's pretraining phase and recognizing the redundancy involved in fine-tuning the entire parameter space, LoRA achieves effective task adaptation with minimal performance degradation. Empirical results demonstrate that LoRA can achieve competitive performance compared to full fine-tuning while requiring only a fraction of the computational resources.

Mathematically, for a forward pass represented as $\mathbf{Y} = \mathbf{XW}$, where $\mathbf{X} \in \mathbb{R}^{h \times d}$ and $\mathbf{W} \in \mathbb{R}^{d \times d}$, LoRA computes the output as follows:

$$\mathbf{Y} = \mathbf{X}\left(\mathbf{W} + \mathbf{L}_a\mathbf{L}_b\right), \tag{1}$$

where $\mathbf{L}_a \in \mathbb{R}^{d \times r}$ and $\mathbf{L}_b \in \mathbb{R}^{r \times k}$. Here, $r$ represents the rank of the adaptation, which can be adjusted to balance efficiency and performance.

QLoRA extends the principles of LoRA by integrating quantization into the adaptation process. Quantization reduces the precision of model parameters, typically converting them from 32-bit floating-point representations to lower precision formats such as 8-bit or even 4-bit integers. This

not only decreases the model's memory footprint but also accelerates inference times, making it particularly beneficial for deployment in resource-constrained environments.

By combining quantization with low-rank adaptation, QLoRA maintains the efficiency advantages of LoRA while enhancing the model's adaptability. Despite the reduction in parameter precision, QLoRA has been shown to preserve high performance levels across various tasks. The QLoRA function for a single linear layer with a single LoRA adapter is defined as follows:

$$\mathbf{Y}^{\text{BF16}} = \mathbf{X}^{\text{BF16}} \left( \text{DQ}(c_1^{\text{FP32}}, c_2^{\text{k-bit}}, \mathbf{W}^{\text{NF4}}) + \mathbf{L}_a^{\text{BF16}} \mathbf{L}_b^{\text{BF16}} \right), \tag{2}$$

where $\text{DQ}(\cdot)$ is the double dequantization function that converts QLoRA's compressed weights from NF4 (Narrow Float 4) format to BF16 (BFloat16) format. This innovative integration allows QLoRA to leverage the strengths of both low-rank adaptation and quantization, offering a robust solution for efficient fine-tuning of LLMs while ensuring performance remains largely unaffected.

## 3.2 H-QLoRA: Hierarchical Quantized Low-Rank Adaptation

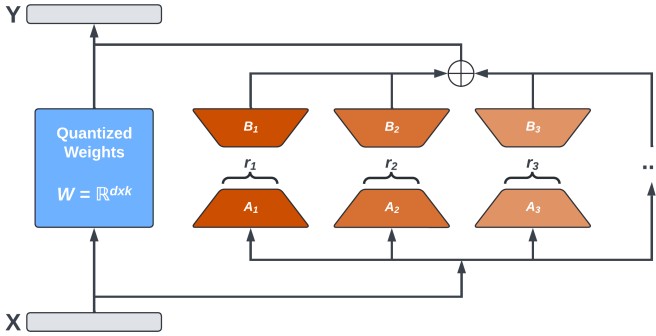

Figure 1: H-QLoRA hierarchical adaptors applied in a forward step. Each LoRA adapter $\mathbf{L}_i$ is comprised of two low rank adapters $\mathbf{L}_{a_i}$ and $\mathbf{L}_{b_i}$ with rank $r_i$. In the figure, the adapters are denoted as $A_i$ and $B_i$.

H-QLoRA introduces a straightforward yet effective implementation strategy that enhances model fine-tuning and performance.

**H-QLoRA Layers:** The implementation leverages direct modifications to the Hugging Face Transformers and PEFT libraries, allowing H-QLoRA to work seamlessly alongside QLoRA. Our approach specifically targets the same classes and functions within these libraries. We inject hierarchical LoRA adapters (an H-QLoRA adapter) in the query (q), key (k), value (v), gate, and both up and down projection layers. Within each instance, the layer is comprised of a series of LoRA $A$ matrices, $B$ matrices, and scalars represented as 1D arrays.

**Forward Pass:** During training, each layer executes a forward pass where the output vectors of each successive adapter layer are cumulatively added. In contrast, at inference time, the adapter layers are summed prior to executing the forward pass, which optimizes runtime efficiency. For a forward pass step described as $\mathbf{Y} = \mathbf{X}\mathbf{W}$, where $\mathbf{X} \in \mathbb{R}^{h \times d}$ and $\mathbf{W} \in \mathbb{R}^{d \times k}$, H-QLoRA computes the outputs as follows:

**Training Time:**

$$\mathbf{Y} = \mathbf{X} \left( \mathbf{W} + \mathbf{L}_{a_1} \mathbf{L}_{b_1} \right) + \mathbf{X} \left( \mathbf{W} + \mathbf{L}_{a_2} \mathbf{L}_{b_2} \right) + \mathbf{X} \left( \mathbf{W} + \mathbf{L}_{a_3} \mathbf{L}_{b_3} \right) + \dots, \tag{3}$$

**Inference Time:**

$$\mathbf{Y} = \mathbf{X} \left( \mathbf{W} + \left( \mathbf{L}_{a_1} \mathbf{L}_{b_1} + \mathbf{L}_{a_2} \mathbf{L}_{b_2} + \mathbf{L}_{a_3} \mathbf{L}_{b_3} + \dots \right) \right), \tag{4}$$

where $\mathbf{L}_a \in \mathbb{R}^{d \times r}$ and $\mathbf{L}_b \in \mathbb{R}^{r \times k}$ represent the LoRA matrices for each adapter layer.

**Advantages of H-QLoRA over QLoRA:** One of the primary advantages of H-QLoRA is its ability to maintain a high level of efficiency while enhancing model performance, especially in smaller and older architectures. The hierarchical structure allows for a more granular adaptation of model parameters, facilitating improved fine-tuning outcomes without increasing the number of trainable parameters. By employing multiple adapter layers in a hierarchical manner, H-QLoRA effectively captures complex relationships and nuances in the data, which can lead to superior model performance compared to traditional QLoRA implementations.

Additionally, the efficiency in inference time achieved by pre-aggregating adapter outputs reduces computational overhead, enabling faster model responses in real-time applications. This makes H-QLoRA particularly suitable for scenarios where both model performance and computational resources are critical, paving the way for broader applicability across various domains.

Overall, H-QLoRA not only simplifies the implementation process but also enhances the adaptability and performance of LLMs, marking a significant advancement over existing methods like QLoRA.

## 4 EXPERIMENT

We investigated the effectiveness of H-QLoRA by fine-tuning the LLaMA and Gemma model families on the OASST1 and Alpaca datasets (Köpf et al., 2023), adhering to the experimental setup established in the QLoRA paper. For the OASST1 dataset, we conducted fine-tuning over 1,875 steps, while for the Alpaca dataset, we utilized 10,000 steps. To evaluate the performance of our fine-tuned models, we employed the MMLU benchmark (Hendrycks et al., 2021). This benchmark assesses the model's capabilities across a diverse array of language understanding tasks, offering a comprehensive measure of its overall effectiveness. We compared the mean 5-shot MMLU accuracy across various hierarchical adapter configurations (H-QLoRA) to develop a detailed performance profile of our enhancements. Our experiments were performed on NVIDIA A6000 GPUs, each posessing 48 GB of memory.

### 4.1 EVALUATION DATASET

The OpenAssistant Conversations Dataset (OASST1) is a comprehensive dataset specifically designed to fine-tune large language models for improved performance in conversational contexts (Köpf et al., 2023). This dataset comprises a wide range of annotated conversations collected from diverse sources, ensuring broad coverage across various domains, topics, and dialogue styles.

OASST1 features high-quality annotations that enhance natural language understanding and generation capabilities, making it particularly suitable for training models intended for interactive and conversational tasks. The conversations within the dataset include multiple turn dialogues, questions and answers, and varied interaction types, which contribute to its richness and versatility.

The Alpaca dataset comprises 52,000 instruction and demonstration examples generated by OpenAI's text-davinci-003 model, specifically designed to enhance instruction-tuning for language models. This dataset builds upon the Self-Instruct framework and incorporates several key modifications to improve data quality, diversity and efficiency. The Alpaca dataset serves as a valuable resource for fine-tuning models to better respond to diverse instructional queries.

As demonstrated by QLoRA, fine-tuning on a compact yet high-quality dataset like OASST1 can yield exceptional performance improvements, even when applied to smaller models. This highlights the dataset's effectiveness in enhancing the conversational abilities of language models, paving the way for more engaging and context-aware interactions.

### 4.2 EVALUATION METRICS

The Massive Multitask Language Understanding (MMLU) benchmark is a comprehensive evaluation framework designed to rigorously assess the capabilities of language models across a diverse array of tasks and domains (Hendrycks et al., 2021). The benchmark encompasses 57 distinct tasks that cover a wide range of fields, including the humanities, STEM (science, technology, engineering, and mathematics), social sciences, and professional domains such as law and medicine.

This diversity allows MMLU to provide a thorough evaluation of a model's general language understanding abilities and domain-specific knowledge. Each task is formulated to test not only the model's proficiency in understanding and generating coherent text but also its capacity to apply reasoning, critical thinking, and knowledge across multiple disciplines.

MMLU is particularly challenging due to its emphasis on high-level cognitive skills, requiring models to demonstrate a nuanced grasp of context and the ability to synthesize information. As a widely recognized standard in natural language processing (NLP), MMLU serves as an invaluable tool for evaluating the performance of models on a broad spectrum of real-world tasks, offering insights into their robustness and applicability in practical scenarios.

### 4.3 H-QLoRA Evaluation Matrix

Our experimental evaluation is designed to comprehensively assess the performance of H-QLoRA across various hierarchical adaptor configurations and hyperparameters. The baseline QLoRA runs are marked in the 64 (baseline) runs). The sum of hierarchical adaptors is always equal to 64 to ensure no increase in trainable parameters

| Mean 5-shot MMLU Accuracy | | | |
|---|---|---|---|
| Model | | Finetuning Method | |
| | Baseline | QLoRA | H-QLoRA |
| Llama 7B | 33.291 | 34.172 | **36.460** |
| Llama-2 7B | 44.391 | 45.286 | **46.253** |
| Llama-3 8B | 61.276 | **62.927** | 62.526 |
| Gemma 2B | 35.825 | 37.998 | **38.295** |
| Gemma 7B | 60.875 | **62.899** | 59.768 |

Table 1: Fine-Tuned LLaMA and Gemma models using the OpenAssistant Conversations Dataset (OASST1).

### 4.4 Memory and Runtime

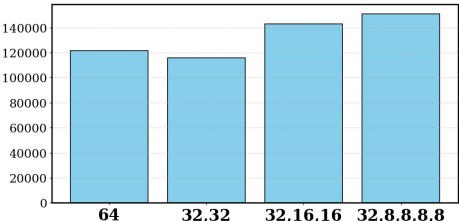

Figure 2: Training runtime (seconds) of Llama 7B on OASST1 with baseline QLoRA (64) and H-QLoRA configurations

**Training:** The H-QLoRA configuration $32, 8, 8, 8, 8$ is around 24.3% slower during training time. The series of adapters, which results in additional computations for longer configurations such as $32, 8, 8, 8, 8$, inevitably slows down the training relative to a singular adapter. We argue that this decline in performance is still comparable to the standard LoRA configuration.

**Memory:** The memory cost of H-QLoRA comparable relative to traditional LoRA fine-tuning because it retains the same number of trainable parameters (additional adapter parameters). This enables the same memory advantage of low-rank adaptation that enables these quantized models to be trained on limited GPU resources.

## 4.5 RESULT

As seen in Table 1, the H-QLoRA config $32, 8, 8, 8, 8$ outperforms baseline rank $64$ fine-tuning for Llama 7B, Llama-2 7B, and Gemma 2B. We also see that for some of the newer or larger models (Gemma 7B, Llamma-3 8B), H-QLoRA is less performant. For Gemma 7B, we observed no significant improvement for any fine-tuning, including the baseline, indicating that the dataset was not optimal for this specific model and benchmark.

## 4.6 ABLATION STUDY

We tested a variety of H-QLoRA configs to determine optimal configurations. We recorded the results of this study in Table 2, which displays the 5-shot MMLU results after fine-tuning various Llama and Gemma models. Within each model study, we included a baseline QLoRA $r = 64$ run. We found that the adapter rank configuration $32, 8, 8, 8, 8$ performs the best in the cases where H-QLoRA outperforms the baseline QLoRA when training on the OASST1 dataset. As a result, we opt to use this H-QLoRA configuration for our adapters for comparative experiments. When training on the Alpaca dataset, the $32, 32$ and $32, 16, 16$ adapters perform best a majority of the time. This highlights a performance variation for the dataset size and quality.

| Mean 5-shot MMLU Accuracy | | | |
|---|---|---|---|
| Model | QLora/H-QLoRA Config | Dataset | |
| | | OASST1 | Alpaca |
| Llama 7B | 64 | 34.172 | 33.804 |
| | 32,32 | 35.278 | **34.398** |
| | 32,16,16 | 36.147 | **34.374** |
| | 32,8,8,8,8 | **36.460** | 32.830 |
| Llama-2 7B | 64 (baseline) | 45.286 | 45.948 |
| | 32,32 | 45.125 | 45.795 |
| | 32,16,16 | 45.913 | **47.469** |
| | 32,8,8,8,8 | **46.253** | 45.647 |
| Llama-3 8B | 64 | 62.927 | 63.293 |
| | 32,32 | **63.092** | **63.691** |
| | 32,16,16 | 62.487 | 63.021 |
| | 32,8,8,8,8 | 62.526 | 61.304 |
| Gemma 2B | 64 | 37.998 | 37.185 |
| | 32,32 | 37.718 | **37.638** |
| | 32,16,16 | 37.271 | 36.326 |
| | 32,8,8,8,8 | **38.295** | 37.089 |
| Gemma 7B | 64 | **62.899** | **53.973** |
| | 32,32 | 61.397 | 47.532 |
| | 32,16,16 | 60.553 | 42.730 |
| | 32,8,8,8,8 | 59.768 | 31.470 |

Table 2: Fine-Tuned LLaMA and Gemma models using OpenAssistant Conversations Dataset (OASST1) and Alpaca.

## 5 DISCUSSION

H-QLoRA presents a novel way to fine-tune quantized large language models built off of the advancements of low-rank adaptation and quantization techniques. By implementing a hierarchical adapter framework, H-QLoRA allows each layer to learn different ranges of data, which is especially useful in quantized models, where high-frequency data becomes degraded. The experimental results show model performance without any significant memory increase for models like Llama 7B, Llama-2 7B, and Gemma 2B.

At the same time, the hierarchical structure introduces some computational overhead during training, especially for the longer H-QLoRA configurations tested. This highlights the slight trade-off between performance gains and practical considerations such as resource availability. Despite this, H-QLoRA has minimal additional memory costs and comparable inference times from pre-aggregating outputs.

Our experimental results show that H-QLoRA performs consistently well with certain models and configuration, but not universally so. It does not consistently outperform traditional methods for all models, such as newer or larger ones such as Llama-3 8B and Gemma 7-B. These findings emphasize considerations such as model architecture, as well as dataset suitability and size. Experiments with the Alpaca dataset show that H-QLoRA struggles against traditional fine-tuning methods.

## 6 CONCLUSION

H-QLoRA offers a significant advancement in the fine-tuning of large language models, effectively addressing the challenges posed by resource constraints. By introducing hierarchical adaptors with low-rank weights, H-QLoRA not only mitigates the accuracy loss associated with aggressive weight quantization but also enhances the model's ability to capture high-frequency information. Our evaluation of H-QLoRA across various instruction datasets demonstrates its superiority over existing methods, proving that it can maintain efficiency while achieving competitive performance. In the future, we plan to investigate the effectiveness of H-QLoRA on additional datasets and architectures, extending beyond natural language processing to explore its broader applicability.

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
