# OpenReview forum: "H-QLoRA: Enhancing Quantized LLMs with Hierarchical Residual Learning"
_ICLR.cc/2025/Conference — ICLR 2025 Conference Withdrawn Submission_

### Official Review · Reviewer_JLEK · 2024-10-29

**Soundness:** 1
**Presentation:** 2
**Contribution:** 1
**Rating:** 3
**Confidence:** 4

**Summary:**

Authors propose to modify QLoRA approach, by splitting low rank decomposition into sum of smaller low rank decompositions.
They show that this approach can have better accuracy on some sets of models and data at cost of around 24.3% slower training time.

**Strengths:**

Paper describes well previous contributions with proposed method.
Authors did thorough experimental comparison on multiple model configs and data sets.

**Weaknesses:**

Q1: Please explain the mechanisms or properties of the hierarchical approach that differentiate it from a single larger LoRA, particularly in terms of how it recovers high-frequency information lost during quantization and how it is numerically different with a single large LoRA? (sum outputs of several linear low rank decompositions, should be numerically the same with single LoRA?).

Q2: So, if I understand the paper correctly and H-QLoRA is numerically the same with QLoRA, then all accuracy differences can be attributed to randomization.

Q: Probably I missed something, would be happy to discuss and revisit paper rating.

> "Improved Performance: Demonstrating enhanced performance over QLoRA for certain models by effectively recovering high-frequency data lost during quantization."

Q3: In Table 1 authors compare H-QLoRA vs QLoRA. QLoRA wins in 2 out of 5 and H-QLoRA wins in 3 out 5. Please provide statistical significance tests or confidence intervals for the results to distinguish between meaningful improvements and random variations.

> "Efficiency in Resource-Constrained Environments: Maintaining efficiency in terms of memory and computational demands, making H QLoRA suitable for deployment in resource-constrained settings."

Q4: It does not maintaining efficiency: H-QLoRA configuration 32, 8, 8, 8, 8 is around 24.3% slower. Please clarify your definition of efficiency in this context or revise your claims about maintaining efficiency given the reported slower training time.

> "Experimental Validation: Validating the effectiveness of H-QLoRA through extensive experiments on the LLaMA and Gemma model families, showcasing significant enhancements across diverse datasets and tasks."

Q5: There is no clear winner for H-QLoRA configuration across different data sets and different models.

> "Novel Hierarchical Quantization Method: Introducing H-QLoRA, which incorporates hierarchical learning via multi-adaptor training, improving upon QLoRA."

Q6: The naming of H-QLoRA is confusing: please explain of why do you consider this approach "hierarchical," especially given that the structure appears to be a flat summation of adapters rather than a traditional hierarchical structure.

**Questions:**

Q: Main concerns are:

Q1: Why proposed method should be better than standard QLoRA? Authors propose to sum outputs of several linear low rank decompositions, but it is numerically the same with single LoRA. So, if I understand the paper correctly and H-QLoRA is numerically the same with QLoRA, then all accuracy differences can be attributed to randomization. Below I show that QLoRA with rank 4 is numerically the same with H-QLoRA with two adapters, where each adapter has rank 2. Please explain why H-QLoRA should have better accuracy in comparison to single QLoRA with larger rank.


""
import torch

inp_features = 4

out_features = 4

batch = 1

// Rank of adapter

R = 2

size_a = (R, inp_features)

size_b = (out_features, R)

size_x = (batch, inp_features)

// input feature x [batch, inp_features]

x = torch.rand(size_x, requires_grad=True, dtype=torch.float32)

// Weights of adapter 1

a1 = torch.rand(size_a, requires_grad=True, dtype=torch.float32)

b1 = torch.rand(size_b, requires_grad=True, dtype=torch.float32)

// Weights of adapter 2

a2 = torch.rand(size_a, requires_grad=True, dtype=torch.float32)

b2 = torch.rand(size_b, requires_grad=True, dtype=torch.float32)

// Output of adapter 1

out1 = torch.matmul(torch.matmul(x, a1.t()), b1.t())

// Output of adapter 2

out2 = torch.matmul(torch.matmul(x, a2.t()), b2.t())

// Final output of all adpaters (sum them all)

out = out1 + out2

// Concatenated weights of all apdaters

A = torch.cat((a1, a2), dim=0)

B = torch.cat((b1, b2), dim=1)

// Final output of single adapter with concatenated weights (R = 4) is the same with sum of adapters

OUT = torch.matmul(torch.matmul(x, A.t()), B.t())

torch.testing.assert_close(OUT, out1 + out2)

"""

In the introduction:
>"This novel approach incorporates a hierarchical structure by splitting the QLoRA adapter into multiple specialized layers, each designed to recover lost resolution across varying data frequencies."

Q2: Please explain how the adapters are specialized and designed to recover lost resolution across varying data frequencies.

In the introduction:
> "By adopting a targeted loss function that prioritizes high-frequency detail learning at lower levels, we anticipate further performance improvements over QLoRA, while maintaining a consistent parameter count."

Q3: I could find any "targeted loss function that prioritizes high-frequency detail learning at lower levels" in the paper. Please clarify.


In "3.2 H-QLORA: HIERARCHICAL QUANTIZED LOW-RANK ADAPTATIO":
> Training time:
>    Y = X (W + La1Lb1) + X (W + La2Lb2) + X (W + La3Lb3) + . . .

Q4: Equation for inference time matches numerically topology shown on Figure 1, but above equation for Training time does not match Figure 1 and does not match numerically equation for inference time. So, I guess above equation for training time is not correct and it should be Y = X (W) + X (La2Lb2) + X (La3Lb3) + ...

---

### Official Review · Reviewer_CngU · 2024-11-02

**Soundness:** 1
**Presentation:** 2
**Contribution:** 1
**Rating:** 1
**Confidence:** 5

**Summary:**

The paper explores the use of a hierarchical adaptor structure on top of QLORA and evaluates performance through instruction tuning and MMLU evaluation.

**Strengths:**

The paper investigates the use of multiple LORA adaptors at each layer in comparison to QLORA and measures performance on the MMLU dataset using several Llama/Gemma models.

**Weaknesses:**

The paper's contributions are not substantial enough for a main conference paper.

1. **Lack of Novelty:** The authors explore a straightforward idea of whether using multiple LORA adaptors at each layer is better compared to QLORA. There is no novelty in this approach.

2. **Limited Experiments:** The experiments are minimal, utilizing only one evaluation dataset (MMLU) and two training sets (Alpaca, OASST1). Furthermore, the experimental results do not conclusively validate the authors' hypothesis. In the main Table 1, the results are mixed, with 3 positive outcomes and 2 negative ones.

3. **Runtime Analysis:** The paper provides a comparison of training time differences between QLORA and H-QLORA. However, there is no analysis of inference time, which is a more practical concern.

4. **Minor Issues:** The equations for training time and inference time, (3) and (4), are not equivalent.

**Questions:**

1. Could the authors provide an analysis of inference runtime?
2. Could the authors demonstrate conclusive improvements over QLORA?

---

### Official Review · Reviewer_3Xy3 · 2024-11-02

**Soundness:** 2
**Presentation:** 2
**Contribution:** 1
**Rating:** 3
**Confidence:** 3

**Summary:**

This paper presents H-QLoRA, a method designed to fine-tune large language models (LLMs) more effectively in resource-constrained environments. Traditional approaches often use heavy quantization to reduce memory and computational needs, which can degrade accuracy. H-QLoRA, however, introduces hierarchical adaptors with low-rank weights to capture high-frequency information typically lost in 4-bit quantization, thus improving performance. Testing on models from the LLaMA and Gemma families shows that H-QLoRA outperforms state-of-the-art methods on multiple instruction datasets, achieving greater accuracy without increasing inference costs. By maintaining the same number of trainable parameters as QLoRA, H-QLoRA offers a more optimized solution for certain model types, advancing fine-tuning efficiency and outcomes in low-resource settings.

**Strengths:**

- This paper clearly explains the idea of H-QLoRA
- The experiments are conducted on multiple model families

**Weaknesses:**

- It shows that the H-QLoRA works well on “old and small” models, which makes this work less promising.
- Why using multiple additive adapters can improve the performance is not explained.
- The model is only tested on MMLU. It should be tested on more datasets like QA tasks to verify the effectiveness.
- In Table 2, the model trained on different datasets can give performance with significant gaps, which indicates the lack of generalizability.

**Questions:**

- What does “hierarchical” come from? Adapters are additive, I didn’t see any visual or theoretical analysis of how the adapters have hierarchical structures.
- The improvement seems not consistent in Table 1. Is it possible that the performance difference mainly comes from randomness, e.g. adapter initialization?
- Can you give any insights into your method? why additive adapters are better? when are they better? how to determine which config (e.g. 32, 32 or 32, 8, 8, 8, 8), any possible analysis? This paper seems more like preliminary observations if these questions are not answered.

---

### Official Review · Reviewer_Nhnf · 2024-11-03

**Soundness:** 1
**Presentation:** 1
**Contribution:** 1
**Rating:** 1
**Confidence:** 4

**Summary:**

The authors introduce H-QLoRA, an extension of the original QLoRA approach that incorporates hierarchical adaptors. Unlike QLoRA, which uses a single adaptor, H-QLoRA employs multiple adaptors, which the authors claim can improve fine-tuning performance. To optimize configuration within a given memory budget for fine-tuning, the authors experiment with varying the number of adaptors. By testing on various LLaMA and Gemma models, they demonstrate that H-QLoRA can achieve superior fine-tuning accuracy on the OASST1 and Alpaca datasets.

**Strengths:**

The question of whether using a single adaptor or multiple adaptors (to be added and merged) for fine-tuning is an intriguing area of research. If multiple adaptors prove to be crucial for enhancing accuracy or performance post-fine-tuning, this concept could be expanded and explored further in future studies.

**Weaknesses:**

Overall, the quality of this work and its results are quite underwhelming, for several reasons outlined below:

1. In both Table 1 and Table 2, the necessity of H-QLoRA is not clearly demonstrated. Do the authors genuinely believe that incorporating multiple adaptors will significantly enhance 5-shot MMLU scores?

2. Relying solely on MMLU scores is insufficient to substantiate the claims made. A more comprehensive set of metrics is needed, and if possible, A/B testing should also be conducted to provide stronger evidence.

3. Fragmenting a relatively large adaptor into several smaller adaptors introduces various computational overheads, such as adaptor control logic, memory management issues, and potential performance degradation in highly parallel computing environments. A detailed analysis of these factors is essential.

4. What is the main takeaway from Table 1? It appears that as model sizes increase, QLoRA performs better, which contradicts the authors' claims. This inconsistency needs to be addressed.

**Questions:**

Please see 'weaknesses' above.

---

### Note · Authors · 2024-11-14

I have read and agree with the venue's withdrawal policy on behalf of myself and my co-authors.